# Learning 3D Medical Image Models From Brain Functional Connectivity Network Supervision For Mental Disorder Diagnosis

## Abstract

For mental disorder diagnosis, most previous works are task-specific and focus primarily on functional connectivity network (FCN) derived from functional MRI (fMRI) data. However, the high cost of fMRI acquisition limits its practicality in real-world clinical settings. Meanwhile, the more easily obtainable 3D T1-weighted (T1w) MRI, which captures brain anatomy, is ofen overlooked in standard diagnostic processes of mental disorders. To address these two issues, we propose CINP (**C**ontrastive **I**mage-**N**etwork **P**re-training), a framework that employs contrastive learning between 3D T1w MRI and FCNs. CINP aims to learn a joint latent semantic space that integrates complementary information from both functional and structural perspective. During pre-training, we incorporate masked image modeling loss and network-image matching loss to enhance visual representation learning and modality alignment. Furthermore, thanks to contrastive pre-training which facilitates knowledge transfer from FCN to T1w MRI, we introduce network prompting. This protocol leverages 3D T1w MRI from suspected patients and FCNs from confirmed patients for differential diagnosis of mental disorders. Extensive experiments across three mental disorder diagnosis tasks demonstrate the competitive performance of CINP, using both linear probing and network prompting, compared with FCN-based methods and self-supervised pre-training methods. These results highlight the potential of CINP to enhance diagnostic processes with the aid of 3D T1w MRI in real-world clinical scenario.

## 1 Introduction

Functional magnetic resonance imaging (fMRI) is a modern neuroimaging technique, which can non-invasively investigate brain function by detecting the blood-oxygen-level-dependent (BOLD) responses to task-related neuronal activity (Logothetis, 2008). Over recent years, there has been growing evidence that mental disorders arise from dysfunction of interconnected patterns of regions-of-interest (ROIs) in the whole brain (Krishna et al., 2023), and fMRI-derived functional connectivity network (FCN), as a graph/network architecture with nodes being ROIs and each edge being functional connectivity (FC) between paired ROIs, has received considerable attention in diagnosis of mental disorders (Yang et al., 2021; Bastos & Schoffelen, 2016), where FC is in general measured as statistical dependence between BOLD time courses of paired ROIs.

A significant amount of work has been dedicated to learning deep and differentiable representations for FCNs, with explorations based on deep learning models, such as graph neural networks (GNN) (Li et al., 2021b; Kan et al., 2022a; Kim et al., 2021; Wang et al., 2024), convolutional neural networks (CNN) (Kawahara et al., 2017; Huang et al., 2017), and graph transformer (Kan et al., 2022b). Despite significant progress, deep learning based FCN analysis for mental disorder diagnosis has yet to be widely adopted in real-wrold clinical practice. There are two pervasive challenges, i.e., the limited generalizability and adaptability caused by the small size of annotated fMRI dataset and the lack of integration with anatomical information/features from the easily obtainable 3D T1w MRI. Mental disorders can remodel functional neural circuitry as mentioned above, and yet the anatomical knowledge of the brain necessarily constrains its function (Pang et al., 2023). This indicates that 3D T1w MRI, which assesses brain anatomy structures, holds potential for mental disorder diagnosis, and

an efficient integration of structural and functional perspectives may provide a more comprehensive view of neurobiological abnormalities in mental disorders, resulting in better diagnostic precision.

It is well known that supervised models usually require sufficient labeled training data, which are expensive and time-consuming. Therefore, self-supervised pre-training (SSP) has recently emerged for establishing a foundation model that can be transferred to downstream biomedical tasks including disease diagnosis and medical image segmentation. For example, considering that MAE (masked autoencoder)-based methods (He et al., 2022; Chen et al., 2023a; Wang et al., 2023b) have achieved high performance in various natural image analysis tasks, masked image modeling (MIM) has been applied to learning robust representations of 3D medical images (Zhou et al., 2023; Chen et al., 2023c) through reconstructing 3D volumes of medical images from their highly-masked versions.

In addition, motivated by visual-language pre-training (VLP) that has received tremendous success on downstream vision and language tasks by pre-training a visual language model on large-scale image-text pairs (Radford et al., 2021; Li et al., 2021a; 2022), efforts have been attempted to adapt self-supervised VLP in the biomedical domain, which means pre-training a visual language model using 3D medical images with natural language supervision from the corresponding radiology reports (Boecking et al., 2022; Bannur et al., 2023). Beyond images and text, it is noteworthy that various modalities of medical imaging can inherently offer contrasting perspectives, such as sMRI and fMRI, which provide structural and functional information of the human brain, respectively. A bidirectional mapping scheme Ye et al. (2023) was proposed to conduct ROI-level contrastive learning between diffusion MRI (dMRI) derived structural connectivity network and BOLD signal. However, the model architecture and the limited number of dMRI data limit the scalability.

In this regard, this paper focuses on self-supervised contrastive pre-training on paired 3D T1w MRI image and FCN data for mental disorder diagnosis. We propose Contrastive Image-Network Pre-training (CINP) to jointly learn 3D T1w MRI and FCN representations that can be fully leveraged by downstream mental disorder diagnosis, as illustrated in Fig. 1. Specifically, paired 3D T1w MRI image and FCN data are fed into a visual encoder and a network encoder to extract the embeddings, respectively. Subsequently, the similarity matrix between image embeddings and FCN embeddings is computed by utilizing cosine similarity to facilitate image-network contrastive loss calculation. Masked image modeling loss and image-network matching loss are involved for better visual representation learning and modality alignment. In particular, we develop network prompting protocol which leverages 3D T1w MRI from suspected patients and FCNs from confirmed patients for differential diagnosis of mental disorders. Networks of different subject classes are used as prompt where the similarity between the embedding of 3D T1 MRI and the embedding of networks is calculated and the image is assigned to the class where the corresponding networks hold the highest similarity with the image. The effectiveness of CINP is finally demonstrated on three mental disorder datasets by comparing CINP with FCN-based models and SSP-based models with linear probing, and network prompting protocols.

To sum up, our contributions are as follows: (1) We proposed CINP which builds a joint latent semantic space for 3D T1w MRI images and FCNs. It can combine the complimentary information from multi-modal neuroimaging and benefit downstream biomedical tasks. (2) We introduced network prompting protocol which can achieve differential diagnosis of mental disorders based on 3D T1w MRI from suspected patients and FCNs from confirmed patients (3) Extensive experiments across three mental disorder diagnosis tasks demonstrate the competitive performance of CINP.

## 2 METHODOLOGY

In this section, we first introduce the architecture of CINP (shown in Fig. 1) in Section 2.1. Then, we present the objective functions used in pre-training of CINP in Section 2.2. Lastly, in Section 2.3, we describe network prompting in detail.

### 2.1 ARCHITECTURE OF CINP

As illustrated in Fig. 1, CINP is mainly composed of a visual encoder, a visual decoder, and a network encoder. For the visual encoder, an 8-layer 3D swin transformer is constructed, which is initialized with weights pre-trained on 5050 3D CT images from (Tang et al., 2022). Through the visual encoder, an input 3D T1w MRI image $I$ is encoded into a feature map with a dimension of $[3, 3, 3]$, which is

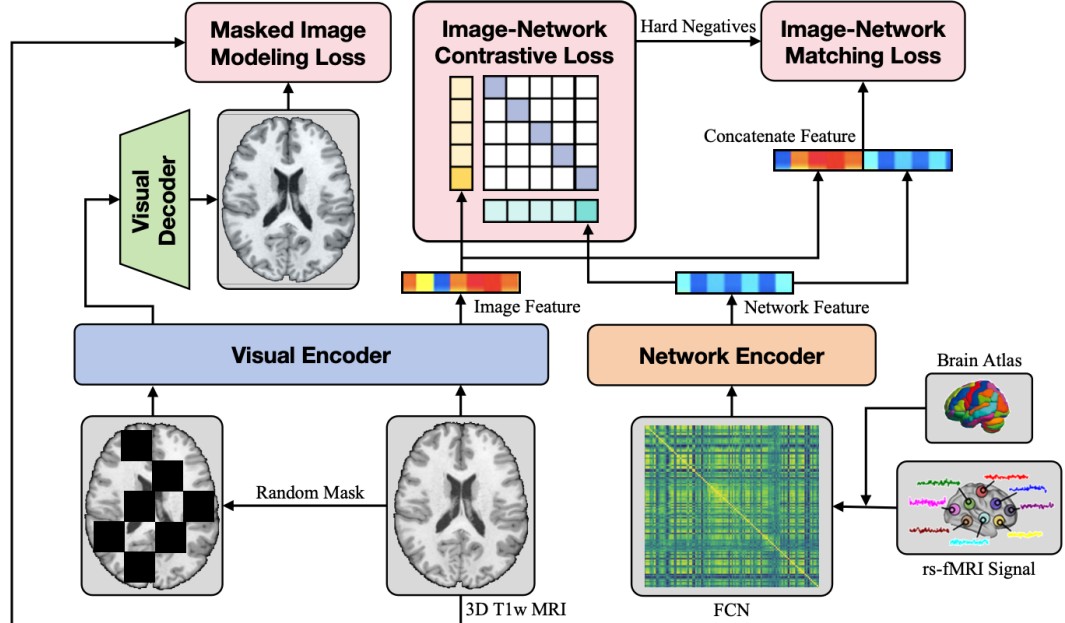

Figure 1: Illustration of CINP. It primarily consists of a visual encoder, a visual decoder, and a network encoder. Image-network contrastive loss, masked image modeling loss, and image-network matching loss are utilized to learn multi-modal interactions between 3D T1w MRI images and FCNs.

then flattened and fed into an average pooling layer and a projection layer to obtain a feature vector $\boldsymbol{v}_I$ with a dimension of 768. The visual decoder based on the transpose convolution layer is the same with that in (Tang et al., 2022), which is utilized to reconstruct masked volumetric medical images for masked imaging modeling (MIM). Due to the outstanding FCN representation capability of brain network transformer (BNT) (Kan et al., 2022b), we adopt BNT as the backbone for the network encoder, where an FCN $N$ is encoded into a 768-dimensional feature vector $\boldsymbol{v}_N$ using transformer blocks, the orthonormal clustering Readout mechanism, and a linear projector.

## 2.2 Pre-training Objectives

Inspired by the insights of previous multi-modal learning methods (Radford et al., 2021; Li et al., 2021a) between image and natural language, CINP is pre-trained with three objective functions associated with image-network contrastive learning (INC), masked image modeling (MIM), and image-network matching (INM), respectively.

**Image-Network Contrastive Learning.** Studies on vision-language pre-training have shown that contrastive learning between image-text pairs can align the hidden semantic spaces of image and text modalities. With this hypothesis, we aim to enhance the representation of 3D medical images with semantic information from FCNs by conducting contrastive learning between image-network pairs. Specifically, Given a pair of image and network data (i.e., a 3D T1w MRI image and an FCN), we want to learn a similarity function $s = \boldsymbol{v}_I^\top \boldsymbol{v}_N$, such that the image-network pair from the same individual have a higher similarity score.

For each image-network pair, the softmax-normalized image-to-network similarity of the $k$-th network $N_k$ and network-to-image similarity of the $k$-th image $I_k$ are respectively calculated as

$$p_k^{\text{in}}(I) = \frac{\exp(s(I, N_k)/\tau)}{\sum_{k=1}^{K} \exp(s(I, N_k)/\tau)} \quad \text{and} \quad p_k^{\text{ni}}(N) = \frac{\exp(s(N, I_k)/\tau)}{\sum_{k=1}^{K} \exp(s(N, I_k)/\tau)}, \tag{1}$$

where the temperature factor $\tau$ is a learnable parameter. Let $\boldsymbol{y}^{\text{in}}(I)$ and $\boldsymbol{y}^{\text{ni}}(N)$ represent the ground-truth similarities of all images and networks, where the similarity of a positive pair (paired image and network data from the same subject) is equal to 1, and the similarity of a negative pair (paired image

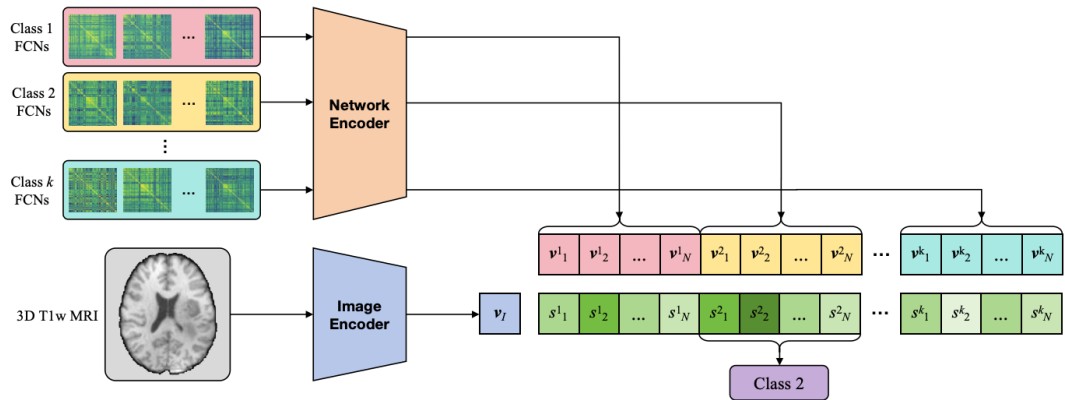

Figure 2: Workflow of network prompting. In the case shown in the figure, the 3D T1w MRI is classified as class 1 for the mean of $s_1^2, s_2^2, \ldots, s_N^2$ is the highest average similarity among all classes.

and network data from different subjects) is equal to $0$. Therefore, based on the cross entropy $H(\cdot, \cdot)$, the INC loss is defined as

$$\mathcal{L}_{\text{INC}} = \frac{1}{2}\mathbb{E}_{(I,N)}\left[H\left(\boldsymbol{y}^{\text{ni}}(I), \boldsymbol{p}^{\text{ni}}(I)\right) + H\left(\boldsymbol{y}^{\text{in}}(N), \boldsymbol{p}^{\text{in}}(N)\right)\right]. \tag{2}$$

**Masked Image Modeling.** Transferred from masked language modeling in natural language processing, masked image modeling (MIM) aims to learn robust representations of images through a self-supervised approach. Given an input 3D medical image $I$, the cutout augmentation masks out square regions in $I$ randomly with volume ratio of $30\%$. The reconstruction output is denoted as $\hat{I}$. The MIM loss is defined by an $L_1$ loss between $I$ and $\hat{I}$, i.e.,

$$\mathcal{L}_{\text{MIM}} = \|I - \hat{I}\|_1 \tag{3}$$

**Image-Network Matching.** Kin to the image-text matching loss, image-network matching (INM) is a binary classification task, which predicts whether a given image-network pair is from the same subject. The concatenation of the image embedding $\boldsymbol{v}_I$ and the network embedding $\boldsymbol{v}_N$ is treated as a joint representation of the image-network pair. Then, $[\boldsymbol{v}_I, \boldsymbol{v}_N]$ is fed into a fully-connected layer followed by softmax to output a binary-class probability $q$. To this end, the INM loss is defined as

$$\mathcal{L}_{\text{INM}} = \mathbb{E}_{(I,N)}\left[H\left(\boldsymbol{z}_{\text{INM}}, \boldsymbol{q}(I, N)\right)\right], \tag{4}$$

where $\boldsymbol{z}_{\text{INM}}$ represents the ground-truth label using a 2-dimensional one-hot vector. Moreover, we sample hard negatives for the INM loss as indicated in ALBEF (Li et al., 2021a). To be specific, in each data batch, the contrastive similarity in (1) is used to sample the hard negative image-network pair which do not come from the same subject but own a high similarity. For each image/network in a mini-batch, one negative network/image is sampled following the distribution of the contrastive similarities, which means that those networks/images having higher similarities to current image/network are more likely to be chosen.

In conclusion, the total pre-training loss of CINP is

$$\mathcal{L} = \mathcal{L}_{\text{INC}} + \alpha\mathcal{L}_{\text{MIM}} + \beta\mathcal{L}_{\text{INM}}. \tag{5}$$

## 2.3 NETWORK PROMPTING

After acquiring the pretrained model, classic approaches employ linear probe or fine-tuning protocols to apply it to downstream tasks. Nonetheless, these methods continue to require a substantial number of annotated data and fall short when encountering the prevalent clinical scenario where fMRI data is not routinely collected for mental disorder diagnosis. To overcome this challenge, as shown in Figure 2, we propose network prompting, which is enlightened from the insight of previous studies (Zhang et al., 2019; Gratton et al., 2018; Alexander-Bloch et al., 2012) that FCNs from

Table 1: Information of datasets used in this study

| Name | Usage | Size | Gender (M/F) | Age (mean±sd) |
|------|-------|------|--------------|---------------|
| HBN (Alexander et al., 2017) | | 2254 | 1455/799 | 10.73±3.39 |
| HCP (Van Essen et al., 2013) | Pre-training | 1080 | 495/585 | (20-40) |
| QTIM (Strike et al., 2023) | | 1024 | 388/636 | 20.71±4.00 |
| CNP (Bilder et al., 2020) | | 261 | 152/109 | 33.29±9.29 |
| ABIDE (Di Martino et al., 2014) | | 855 | 719/136 | 16.92±7.91 |
| ADHD (consortium, 2012) | Evaluation | 872 | 538/334 | 11.98±3.34 |
| SRPBS (Tanaka et al., 2021) | | 1397 | 799/598 | 38.37±13.65 |

various subject classes (e.g., health control and autism spectrum disorder) hold significant differences at the group level, as well as the hypothesis that pre-trained CINP can embed both 3D T1w MRI image and FCN data to a shared semantic space.

Specifically, we first averaged fMRI-derived FCNs of confirmed patients from different subject classes to maintain one or more group-level reference networks which help eliminate subject-level biases. the 3D T1w MRI from the suspected patient waiting for diagonsis and group-level reference networks are encoded to their corresponding visual embedding and network embeddings, respectively. Then, we calculate the similarities between the visual embedding and the network embeddings. According to the average similarities of the visual embedding with each class's FCNs, we assign the image to the classes with the highest average similarity.

## 3 EXPERIMENTS

### 3.1 DATASETS AND PREPROCESSING

**Datasets.** We downloaded several publicly available datasets where the subjects have both 3D T1w MRI images and resting-state fMRI (rs-fMRI) data. As a result, a healthy cohort used for pre-training was composed of four datasets, i.e., HBN (Alexander et al., 2017), HCP (Van Essen et al., 2013), QTIM (Strike et al., 2023), and CNP (Bilder et al., 2020), and three mental disorder datasets (i.e., ABIDE (Di Martino et al., 2014), ADHD (consortium, 2012), and SRPBS (Tanaka et al., 2021)) were used for evaluation. We listed these datasets with basic demographic information in Table 1. Briefly, the pre-training cohort consisted of 4619 healthy subjects. For the evaluation datasets, the ABIDE dataset included 460 health controls (HCs) and 395 patients diagnosed with Autism Spectrum Disorder (ASD); the ADHD dataset included 547 HCs and 325 patients diagnosed with Attention Deficit Hyperactivity Disorder (ADHD); and the SRPBS dataset included 783 HCs and 614 patients individually diagnosed with nine distinct mental disorders, including ASD, Major Depression Disorder (MDD), Obsessive Compulsive Disorder (OCD), schizophrenia spectrum disorder (SSD), and so on.

**Data Preprocessing.** All MR imaging data were preprocessed by fMRIPrep (Esteban et al., 2019), an easily accessible, state-of-the-art fMRI data preprocessing pipeline that is robust against variations in scan acquisition protocols. For 3D T1w MRI images, intensity correction, skull-stripping, and spatial normalization to standard anatomical space (MNI152) were performed. For rs-fMRI data, slice timing correction, head-motion correction, skull-stripping, co-registration to T1 reference, and spatial normalization to MNI152 space were conducted. After such preprocessing procedures, both 3D T1w MRI images and rs-fMRI data had a voxel size of $2mm^3$. To derive FCNs from rs-fMRI on the automated anatomical labelling (AAL) atlas (Tzourio-Mazoyer et al., 2002) that divide the whole brain into 116 ROIs, we defined FC as Pearson's correlation between BOLD time courses of paired ROIs. To align the node feature dimensions of the graph-represented FCNs from datasets where the scanning durations were different, we employed nodal connection profiles, i.e., the corresponding row for each node in the FCN matrix, as the node features. The defined node features have been demonstrated to achieve superior performance over other kinds of node features, such as node identities, degrees, and eigenvector-based embeddings (Cui et al., 2022; Kan et al., 2022a).

## 3.2 BASELINES

**FCN-based Methods.** We deployed four types of FCN-based models based on their network architectures for comparison, which were GNN-based (GCN (Kipf & Welling, 2016) and BrainGNN (Li et al., 2021b)), hyperGNN based (HGCN (Wang et al., 2024) and MHAHGEL (Wang et al., 2024)), CNN-based (BrainNetCNN (Kawahara et al., 2017)), and transformer-based (vanilla Transformer (Kan et al., 2022b) and BrainNetworkTransformer (Kan et al., 2022b)), respectively. It is worth mentioning that we here used a single-atlas version of MHAHGEL. These methods all utilized FCNs as inputs, and were trained from scratch on the three mental disorder datasets separately in that they were task-specific methods. For fair comparison, we followed the hyper-parameter selection and implementation of these methods in the corresponding papers.

**Medical SSP based Methods.** Three medical SSP based methods were used for comparison, which were MedicalNet (Chen et al., 2019), PRCLv2 (Zhou et al., 2023), and Swin-UNETR (Tang et al., 2022), respectively. These baseline models for comparison applied 3D medical images (i.e., computed tomograph (CT) and sMRI) as inputs and pre-training resources, and were executed following the instructions and implementation in their respective papers as well. We employed both linear probe and fine-tuning protocols on these methods. For linear probe, we used the pre-training models provided in their papers. The feature maps of the last layer of these methods were flattened by an average pooling layer and reshaped to feature vectors, which were then directly fed into an SVM classifier for classification. On the other hand, considering the semantic gap and distribution shift between the pre-training data and test data, we also fine-tuned the models that were initialized with pre-trained weights for 10 epochs on the evaluation datasets.

## 3.3 EXPERIMENTAL SETTINGS

**Implementation Details.** The CINP pre-training model was implemented with PyTorch 1.13.1 and MONAI 1.2.0 (Cardoso et al., 2022). For pre-training, we utilized the Adam optimizer with an initial learning rate of $10^{-5}$ and a weight decay of $10^{-5}$. The cosine annealing schedule was applied for the learning rate decreasing to $10^{-6}$. The batch size was set to 256. The pre-training performed on a server equipped with 8 NVIDIA A800-80G GPUs for 400 epochs cost about 100 hours. Since the scales of the three component losses are similar, we set $\alpha = \beta = 1$ in the formula 5.

During the pre-training, the input 3D T1w MRI images were randomly augmented in order to learn robust visual representations. Random augmentations included Gaussian noise addition, random flipping, random intensity scaling, and random intensity shifting. After augmentation, the volumes of 3D T1 sMRI images were resized to $96 \times 96 \times 96$.

**Performance Evaluation.** The diagnoses of ASD and ADHD were two binary classification tasks on ABIDE and ADHD datasets, respectively, which were evaluated in terms of accuracy (ACC), area under ROC curve (AUC), and Matthews correlation coefficient (MCC). For the SRPBS dataset, the classification of ten classes among HC, ASD, MDD, OCD, SSD, pain, bipolar disorder, dysthymia, and other mental disorders was conducted, and ACC and MCC were calculated for evaluation. For linear probe and fine-tuning protocols, we divided the evaluation dataset into training (70%), validation (10%), and testing (20%) sets randomly, ensuring fair comparison of all models. Note that the network prompting protocol had the same data partition, where the FCNs were sampled from the training set and 3D T1w MRI images in the testing set were evaluated.

## 3.4 QUANTITATIVE RESULTS

**Comparision with baselines.** As presented in Table 2, we compared our CINP model using linear probe protocol with FCN-based models and SSP-based models. One can see from Table 2 that the CINP model achieved the best accuracy and AUC on the ADHD dataset, as well as the best accuracy and the second-highest MCC on the SRPBS dataset.

**Evaluation of network prompting protocol.** In Table 3, we evaluate our CINP model using introduced network prompting protocol. the FCNs were sampled from the training set (70% of the evaluated mental disorder dataset), which was treated as previously diagnosed patients. The settings contianed averaged all samples (overlap 100% of data) from different classed (i.e., HC and

Table 2: Classification results of different methods on three mental disorder datasets in terms of ACC (%), AUC (%), and MCC (%). The highest performance is marked in bold, and the second-best is marked using underlines. The values in parentheses are 95% confidence intervals (95% CIs).

| Type | Method | Training | Network | ABIDE | | | ADHD | | | SRPBS | |
|---|---|---|---|---|---|---|---|---|---|---|---|
| | | | | ACC | AUC | MCC | ACC | AUC | MCC | ACC | MCC |
| FCN based | GCN | From scratch | GNN | 61.64 (58.23-63.41) | 64.30 (62.85-67.29) | 21.87 (19.14-23.54) | 60.78 (58.57-62.72) | 59.60 (56.31-61.76) | 13.16 (10.27-15.71) | 53.21 (50.42-55.69) | 7.91 (5.11-9.34) |
| | BrainGNN | | GNN | 55.79 (53.72-57.51) | 58.67 (56.31-60.36) | 9.57 (7.14-11.29) | 57.23 (54.99-58.84) | 55.31 (52.41-56.98) | 12.26 (10.18-13.89) | 53.08 (50.62-55.28) | 18.75 (15.06-20.91) |
| | BrainNetCNN | | CNN | 62.92 (58.16-65.25) | 68.70 (64.54-70.86) | 26.12 (22.39-29.36) | 63.31 (60.40-65.93) | 63.35 (60.87-65.17) | 19.57 (16.35-21.14) | 51.43 (48.91-53.12) | 19.99 (16.12-22.07) |
| | HGCN | | hyperGNN | 62.34 (59.40-64.49) | 66.71 (62.51-68.23) | 13.03 (10.54-15.75) | 62.39 (58.30-64.18) | 62.52 (58.06-64.65) | 18.61 (15.33-20.59) | 54.36 (50.98-56.86) | 18.61 (15.74-20.62) |
| | MHAHGEL | | hyperGNN | 63.51 (61.33-65.49) | 68.09 (66.71-69.97) | 29.19 (27.62-31.88) | 64.11 (62.57-66.22) | 62.00 (60.34-64.15) | 18.41 (16.73-20.61) | 56.58 (54.07-58.74) | 20.22 (18.72-22.27) |
| | vanillaTF | | Transformer | 61.29 (55.59-63.18) | 66.46 (62.34-68.76) | 22.85 (19.75-23.99) | 59.06 (55.46-61.71) | 58.20 (54.43-60.69) | 13.39 (10.53-15.80) | 51.43 (46.18-53.72) | 19.60 (16.12-21.48) |
| | BNT | | Transformer | **65.96** (62.99-67.84) | **72.00** (69.86-73.94) | **31.80** (27.81-33.62) | 63.42 (61.14-64.78) | 64.47 (62.11-65.92) | 19.86 (17.46-21.24) | 57.08 (54.61-59.82) | **29.07** (26.39-31.25) |
| SSP based | MedicalNet | Linear probe | ResNet | 53.92 (52.64-57.31) | 52.37 (50.15-53.82) | 3.76 (1.03-6.02) | 63.54 (60.55-66.86) | 64.74 (59.77-67.11) | 10.25 (6.03-14.38) | 55.69 (53.26-58.20) | 13.32 (8.32-15.48) |
| | PRCLv2 | | ResNet | 55.20 (52.69-58.59) | 51.30 (48.34-55.88) | 8.93 (2.68-12.56) | 66.18 (63.07-69.27) | 67.47 (63.28-70.67) | 22.91 (15.07-29.07) | 56.26 (53.54-59.05) | 13.76 (8.95-15.67) |
| | Swin-UNETR | | SwinViT | 55.79 (52.63-59.06) | 54.17 (53.51-56.33) | 9.50 (8.51-12.79) | 66.63 (63.53-69.84) | 67.58 (64.23-71.21) | 23.72 (16.75-29.57) | 56.33 (53.90-58.98) | 14.39 (12.29-16.12) |
| | MedicalNet | Fine tuning | ResNet | 54.39 (52.95-55.61) | 51.25 (49.98-52.61) | 4.56 (1.87-5.78) | 65.71 (62.32-69.53) | 68.38 (64.22-70.30) | 24.09 (18.09-30.24) | 58.57 (55.97-60.19) | 14.10 (10.28-15.77) |
| | PRCLv2 | | ResNet | 54.60 (50.41-56.62) | 59.65 (58.99-61.82) | 14.19 (11.48-17.80) | 67.82 (66.61-68.69) | 68.23 (66.59-70.60) | 22.90 (22.90-27.49) | 57.50 (53.82-58.98) | 11.74 (8.61-13.26) |
| | Swin-UNETR | | SwinViT | 57.31 (55.46-58.24) | 59.83 (56.61-61.37) | 14.53 (12.93-16.49) | 67.39 (66.17-68.48) | 68.39 (66.74-70.55) | **26.33** (24.41-28.28) | 57.14 (53.54-58.77) | 14.91 (7.26-16.55) |
| - | CINP (Ours) | Linear probe | - | 62.86 (60.16-64.32) | 62.75 (61.04-64.46) | 19.22 (17.35-21.68) | **69.08** (67.17-70.89) | **71.00** (69.58-72.49) | 25.33 (23.37-27.01) | **64.29** (63.02-65.77) | 22.26 (20.78-23.29) |

diseases groups) in the training set to obtain 1group-level reference network, partition all samples (data usage:100%) from different classed into 5/10 subsets, and average each subset (i.e, overlap 20%/10% percents of samples) to obtain 5/10 reference networks, and use 50% of samples from different classed, partition them into 1/5/10 subsets, average each subset (i.e, overlap 50%/10%/5% of samples) to obtain 1/5/10 reference networks.

### 3.5 RESULTS ANALYSIS

**The supervision from FCNs can enhance the predictive power of 3D T1 MRI images for mental disorder identification.**    Compared with the best results among SSP-based models, our CINP model using linear probe improved the accuracy performance by $5.55\%$, $1.26\%$, and $5.72\%$ on the ABIDE, ADHD, and SRPBS datasets, respectively. The AUC performance on the ABIDE and ADHD datasets were respectively improved by $2.92\%$ and $2.61\%$, while the MCC performance on the ABIDE dataset was increased by $4.69\%$. This indicates that by conducting contrastive learning between 3D T1 MRI images and FCNs, their mutually complementary information can be fully captured, benefitting mental disorder identification. Moreover, by fine-tuning the model on downstream tasks, SSP-based models achieved better performance than CINP using linear probe, which means that the performance of CINP holds potential to be further improved.

**The predictive power of various neuroimaging modalities differs across different mental disorders.**    Although the CINP model with the linear probe protocol defeated all SSP-based models and some FCN-based models, it failed to achieve the-state-of-art performance on the ABIDE dataset while FCN-based models were weaker than all SSP-based models and the CINP model on the ADHD dataset. This suggests that 3D T1 MRI images can provide more effective information for the diagnosis of ADHD while the ASD identification may require more involvement of FCNs, even though the imaging feature embeddings were enhanced by FCN supervision during contrastive pre-training. Note that on the SRPBS dataset where multiple mental disorders were included, the performance of FCN-based methods and SSP-based methods were interlaced, and the CINP models which were pre-trained on both 3D T1 MRI images and FCNs achieved the best performance. It again emphasizes the importance of integration of 3D T1 MRI images and FCNs for mental disorder diagnosis.

**The application of network prompting in CINP shows considerable performance.**    Using group-level reference networks, the CINP model with the network prompting protocol held the best MCC performance (29.33%)on the ADHD dataset. Considering that network prompting is a few-shot method, its performance, surpassing all SSP-based methods that used extensive data for fine-tuning on the ABIDE dataset was particularly noteworthy. This demonstrates the feasibility of using a

Table 3: Classification results of different network prompting settings on two mental disorder datasets in terms of ACC (%), AUC (%), and MCC (%). The highest performance is marked in bold, and the second-best is marked using underlines. The values in parentheses are 95% confidence intervals (95% CIs).

| Training Data Usage | Overlap Percent | Network Number | ABIDE | | | ADHD | | |
|---|---|---|---|---|---|---|---|---|
| | | | ACC | AUC | MCC | ACC | AUC | MCC |
| 100% Used | 100% | 1 | 54.39 (50.47-58.64) | 52.77 (48.45-55.93) | 11.32 (5.04-17.19) | 57.65 (54.25-59.67) | 60.98 (57.39-63.92) | 18.80 (15.49-21.11) |
| | 20% | 5 | 55.50 (51.67-59.40) | 53.76 (50.52-57.31) | 13.21 (11.23-15.12) | 58.63 (55.11-61.45) | 62.23 (58.03-64.32) | 17.02 (14.12-19.99) |
| | 10% | 10 | 53.53 (49.95-57.67) | 53.85 (48.40-60.50) | 6.83 (2.45-10.18) | 63.97 (60.26-67.71) | 60.80 (54.31-66.13) | 26.25 (22.93-30.08) |
| 50% Used | 50% | 1 | 59.79 (54.94-64.94) | 57.41 (51.78-62.88) | 19.93 (10.51-28.22) | 58.66 (54.13-62.00) | 57.74 (54.37-61.52) | 17.61 (13.53-21.41) |
| | 10% | 5 | 55.27 (52.71-57.87) | 59.57 (52.63-64.49) | 10.85 (6.46-14.50) | **64.74** (58.26-69.74) | **67.32** (61.37-73.55) | 26.90 (20.32-32.48) |
| | 5% | 10 | 55.15 (51.01-59.84) | 57.09 (54.62-60.16) | 10.01 (8.24-12.30) | 61.61 (58.40-64.91) | 59.75 (56.12-63.15) | 20.45 (17.31-23.06) |
| 10% Used | 10% | 1 | 56.45 (50.70-61.57) | 58.74 (51.04-64.80) | 15.09 (8.29-22.79) | 62.27 (58.57-67.48) | 61.78 (56.34-68.96) | 21.70 (11.10-31.36) |
| | 2% | 5 | **62.56** (58.72-66.04) | **61.34** (57.73-64.97) | **25.84** (21.48-29.01) | 64.15 (59.40-69.28) | 65.33 (60.53-69.26) | 27.11 (23.38-30.65) |
| | 1% | 10 | 57.24 (54.57-60.05) | 56.10 (52.90-60.28) | 13.97 (10.87-17.25) | 64.66 (62.90-69.42) | 63.48 (60.57-66.68) | **29.16** (27.41-31.06) |

Table 4: Classification results of CINP with different combinations of pre-training objective functions on three mental disorder datasets.

| Loss | | | ABIDE | | | ADHD | | | SRPBS | |
|---|---|---|---|---|---|---|---|---|---|---|
| $\mathcal{L}_{INC}$ | $\mathcal{L}_{MIM}$ | $\mathcal{L}_{INM}$ | ACC | AUC | MCC | ACC | AUC | MCC | ACC | MCC |
| ✓ | ✗ | ✗ | 58.57 | 59.94 | 17.94 | 62.86 | 61.20 | 11.68 | 58.93 | 13.64 |
| ✓ | ✓ | ✗ | 60.00 | 57.00 | 16.95 | 66.23 | 68.57 | 20.67 | 60.71 | 17.04 |
| ✓ | ✗ | ✓ | 61.42 | 60.13 | **21.52** | 63.37 | 64.69 | 17.50 | 61.43 | 16.90 |
| ✓ | ✓ | ✓ | **62.86** | **62.75** | 19.22 | **69.08** | **71.00** | **25.33** | **64.29** | **22.26** |

large number of image-network pairs for contrastive pre-training and subsequently leveraging the pre-trained model for mental disorder diagnosis, even when only a limited number of FCNs from diagnosed patients are available.

### 3.6 ABLATION STUDIES

We conducted ablation studies on the CINP models with linear probe, which were pre-trained with different combinations of the three objective functions in Section 2.2. The results were summarized in Table 4. One can see from Table 4 that both the MIM loss and INM loss can benefit the performance of CINP on all the three evaluation datasets, and the usage of all the three objective functions preformed best. To be more specific, on the ABIDE dataset, the MIM loss improved the accuracy by $1.43\%$ and the INM loss improved the accuracy by $2.85\%$, while on the ADHD dataset, the MIM loss improved the accuracy by $3.37\%$ and the INM loss improved the accuracy by $0.51\%$. The MIM loss predominantly enhanced the model's representations of 3D T1 MRI, whereas the INM loss aligned the feature representations of 3D T1 MRI and FCNs. Therefore, from the objective function perspective, this demonstrates again that the predictive power of various neuroimaging modalities differs across different mental disorders.

## 4 RELATED WORK

### 4.1 SELF-SUPERVISED PRE-TRAINING FOR 3D MEDICAL IMAGE ANALYSIS

Self-supervised pre-training for 3D medical image analysis primarily follows two main approaches: contrastive learning and pretext task. Following the approaches on natural images, medical contrastive learning continues to focus on the interplay between medical images and the descriptive

text. Gloria (Huang et al., 2021) was an attention-based framework that contrasted sub-regions of 2D medical images and words in the paired report to learn global and local representations. To enlarge the size of 3D medical image-text pairs, several methods (Liu et al., 2023; Chen et al., 2023b) attempted to convert the 3D images into 2D slices and subsequently employ generative models to generate captions for the 3D medical images. To create unified representations for diverse modalities of medical images, UniMedI (He et al., 2023) which utilized diagnostic reports as common semantic space was proposed. On the other hand, pretext tasks were designed to learn representation from inner structure space of 3D medical images and have shown promise in dense prediction tasks, e.g., lesion segmentation. MedicalNet (Chen et al., 2019) and PCRL (Zhou et al., 2023) were ResNet-based architectures, pre-trained with the segmentation task and the pixel restoration task on 3D medical images, respectively. Pre-training with MIM and ViT-based MED3D (Chen et al., 2023c) showed effectiveness under different practical scenarios. Swin-UNETR (Tang et al., 2022) was a 3D swin transformer with three pretext tasks including rotation prediction, masked volume inpainting, and contrastive coding, while MIS-FM (Wang et al., 2023a) was pre-trained on the self-supervised segmentation task with a large-scale unannotated dataset.

## 4.2 FCN-BASED MENTAL DISORDER DIAGNOSIS

Given that FCNs can be easily represented with graph structures, it is quite natural to apply GNNs, which have the capacity to capture information from neighbor nodes in a neural circuit of a specific brain node, for mental disease identification based on FCNs. BrainGNN (Li et al., 2021b) designed ROI-aware GNNs to leverage the functional information in brain networks and used a special pooling operator to select those crucial nodes. FBNetGen (Kan et al., 2022a) learned to generate the structure of FCNs and explored the explainability of the generated brain networks towards downstream tasks. Without focusing on static brain networks, STAGIN (Kim et al., 2021) utilized GNNs with spatio-temporal attention to model dynamic FCNs, and MHAHGEL (Wang et al., 2024) constructed FCNs as hypergraphs and applied hypergraph convolutional networks for capturing complex relations among multiple brain nodes. Besides, CNN can also be used to facilitate mental disorder diagnosis and brain network analysis, such as BrainNetCNN (Kawahara et al., 2017) and Deep Convolutional Auto-Encoder (Huang et al., 2017). More recently, researchers have shown increasing interest in graph transformer, due to its outstanding performance in graph representation learning while most existing Transformer-based networks (Ying et al., 2021; Kreuzer et al., 2021) have achieved only limited success in FCN analysis. To overcome this limitation, BNT (Kan et al., 2022b) was proposed with orthonormal clustering readout to learn distinguishable cluster-aware node embeddings and informative graph embeddings, which elevated the performance of graph transformer based models over GNN-based models. However, these FCN-based models are limited by the small number of annotated fMRIs and the absent of structural information, which may restrict their performance and generalization abilities.

## 5 CONCLUSION

In this paper, we proposed CINP, a contrastive pre-training framework, on paired 3D T1w MRI image and FCN data for mental disorder diagnosis. The image-network contrastive loss, masked image modeling loss, and network-image matching loss are utilized for jointly learn 3D T1w MRI image and FCN representations that can be fully leveraged to the downstream mental disorder diagnosis task. Thanks to contrastive pre-training which facilitates knowledge transfer from FCN to 3D T1w MRI, we introduce network prompting. This protocol leverages 3D T1w MRI from suspected patients and FCNs from confirmed patients for differential diagnosis of mental disorders by calculating the similarity between the sMRI embedding and the FCN embeddings of different subject classes. Extensive experiment demonstrated the effectiveness of CINP on three downstream mental disorder classification tasks. With pre-trained CINP, 3D T1 MRI which are easily accessible can be utilized for mental disorder detection and eventually benefit clinical practices.

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
