# OpenReview forum: "Learning 3D Medical Image Models From Brain Functional Connectivity Network Supervision For Mental Disorder Diagnosis"
_ICLR.cc/2025/Conference — ICLR 2025 Conference Withdrawn Submission_

### Official Review · Reviewer_Rh9t · 2024-10-30

**Soundness:** 2
**Presentation:** 2
**Contribution:** 2
**Rating:** 5
**Confidence:** 4

**Summary:**

This paper proposed a contrastive image network pre-training method for mental disorder diagnosis. By using contrastive learning between the structure MRI and functional MRI, the proposed can use the useful information from both sMRI and fMRI for mental disorder diagnosis. The proposed method has been compared with several competing methods. The results show that the proposed method achieves better performance than the  competing methods.

**Strengths:**

1) A contrastive image-network pre-training method is proposed to use the multi-modal data for mental disorder diagnosis.
2) The performance shows that the proposed method achieves better performance.

**Weaknesses:**

1) The significant advantage of brain networks lies in their interpretability, yet the paper lacks an interpretability analysis related to diseases.
2）Many multimodal methods based on functional and structural MRI have been proposed, but this paper does not compare with these methods.
3) The details of the methods are not clear enough.

**Questions:**

1) Present findings related to diseases and provide analysis.
2）Compared with the multi-model fusion methods.

---

### Official Review · Reviewer_nAjV · 2024-10-30

**Soundness:** 2
**Presentation:** 3
**Contribution:** 2
**Rating:** 5
**Confidence:** 4

**Summary:**

The paper studies the problem of mental disorder diagnosis using multimodal MRI data. Specifically, it proposes a framework called CINP (Contrastive Image-Network Pre-training), which applies contrastive learning between 3D T1-weighted (T1w) MRI and functional connectivity networks (FCNs) derived from fMRI. CINP aims to create a joint latent space integrating functional and structural information, enhancing diagnostic capabilities. During pre-training, the framework incorporates masked image modeling and network-image matching losses to improve modality alignment and representation quality. Moreover, the CINP contains a network prompting, enables the use of 3D T1w MRI from suspected patients and FCNs from confirmed cases to differentiate mental disorders. Extensive results on public datasets shows CINP has good performance.

**Strengths:**

1. The paper addresses a significant problem in mental disorder diagnosis by leveraging both structural and functional MRI data, a topic relevant and valuable to the ICLR communities.
2. The figures and descriptions in the paper are well-organized and clear, which aids in understanding the proposed approach and the CINP framework.
3. The writing in the paper is clear, making the methodology and results easy to follow.

**Weaknesses:**

1. The paper's technical contributions appear limited, as it mainly combines existing methods like contrastive learning and masked autoencoders (MAE) without significant innovation in methodology. The CINP framework may be seen as a straightforward integration of known techniques rather than a novel approach.
2. There is still room for performance improvement. The performance of CINP on the ABIDE dataset is noticeably lower than that of the baselines, as indicated in Table 2. This suggests that the framework may not be fully optimized or may have limitations, and further improvements are needed to make it competitive across all datasets.
3. The assumption that sMRI and fMRI features can be effectively aligned using contrastive learning lacks theoretical or empirical support from a neuroimaging or neuroscience perspective. This forced alignment may overlook important modality-specific differences, making the approach less effective for capturing unique structural-functional relationships in brain data. (See my questions below.)

**Questions:**

My main concern is with the rationale behind using contrastive loss in CINP to model sMRI and fMRI representations. Based on my knowledge, there is a clear distinction between neuroimaging and image-text domains. In the image-text domain, it makes sense to maximize the differences between different categories while minimizing the differences within the same category. For instance, aligning a cat’s textual description with its image, while creating a contrast with dog-related text and images, is logical. However, in neuroimaging studies, why should we align sMRI and fMRI feature representations in this way? And why should we increase inter-subject differences?

I also note that a healthy cohort was used for CINP’s pre-training. The differences within a healthy population do not fall into distinct categorical boundaries, so it is unclear what meaningful patterns are learned by maximizing inter-subject representation differences. Brain imaging features of healthy individuals are generally quite similar in terms of overall structure and function. By maximizing these differences within healthy individuals, the model may pick up biologically insignificant or irrelevant details, diverting its focus from core features. Such differences are more likely to be noise than meaningful information. In contrastive learning, distinct category boundaries are typically required to generate positive and negative samples, but healthy individuals do not naturally fall into clear categories, making the construction of inter-subject contrasts somewhat artificial.

Additionally, enforcing a “strict alignment” between sMRI and fMRI features may lack generalizability. For mental disorders, patients do not always exhibit abnormalities in both sMRI and fMRI simultaneously. For example, a patient might have a normal gray matter thickness in the prefrontal cortex as shown by sMRI, yet fMRI may reveal weakened functional connectivity between the prefrontal cortex and other regions (such as the parietal lobe or hippocampus). In such cases, a “strict alignment“ seems less appropriate. Similarly, even among healthy individuals, while there may be some correlation between gray matter thickness and functional connectivity, it is not necessarily consistent. Forcing alignment within a healthy cohort may lead the model to mistakenly assume a strong correlation between structural and functional features, thereby overlooking the natural variability and dynamic nature of functional connectivity.

I look forward to seeing the authors' response and would consider raising my score if they can adequately address my concerns.

---

> ### Comment · Reviewer_nAjV · 2024-11-26
>
> Thank you for your response and the clarification. I have updated my score accordingly.

---

> > ### Author Response · Authors · 2024-11-27
> >
> > Thanks for your prompt response despite such a busy period. We deeply appreciate your support. Your valuable and constructive comments have greatly helped us in enhancing our work. We will try our best to keep improving our work. Please feel free to let us know if you need any further clarification or have any other concerns.

---

### Official Review · Reviewer_Ay6P · 2024-10-30

**Soundness:** 2
**Presentation:** 2
**Contribution:** 2
**Rating:** 5
**Confidence:** 4

**Summary:**

The authors propose a contrastive learning approach to learn from both fMRI and anatomical T1w MRI with the aim to differentiate various psychiatric disorders.

**Strengths:**

- The approach appears original in the way it allows the diagnosis of psychiatric disorders based on the patient's T1w MRI only. At the inference stage, two components are required: i) the T1w MR image of the patient to be diagnose and ii) fMRI data of patients with known diagnosis arranged in different diagnostic classes. The predicted diagnosis corresponds to one for which the similarity between the T1 and fMRI embedding was the highest.
- This particular approach seems novel.

**Weaknesses:**

- The paper is not always easy to follow, in particular the Quantitative results section. Careful proofreading would help.
- It is not entirely clear how the ABIDE, ADHD and SRPBS data sets were split and used during the evaluation: i) were the splits stratified, and if so on what criteria, ii) were the results of the ablation study displayed in Table 4 obtained on the test set (and so was hyper-parameter selection based on the test set)?
- The diagnostic classes are not balanced and no metric adapted to this scenario is used to assess the performance.
- The references of the first paragraph of the introduction mostly do not seem appropriate:
    - 'Over recent years, there has been growing evidence that mental disorders arise from dysfunction of interconnected patterns of regions-of-interest (ROIs) in the whole brain (Krishna et al., 2023) […].' This paper is about glioblastoma, it has nothing to do with fMRI nor mental disorders.
    - '[…] fMRI-derived functional connectivity network (FCN) […] has received considerable attention in diagnosis of mental disorders (Yang et al., 2021; Bastos & Schoffelen, 2016) […].' The first paper is about diffusion MRI and the second one describes functional connectivity analysis in general and is not focused at all on mental disorders.
- The organisation of the paper does not seem optimal. I do not see the point of having a Related Work section at the end of the paper knowing that the methods are only briefly describes and no conclusion is being drawn.
- Several works with similar aims should be cited and discussed, e.g.
    - He, Zhibin, et al. "F2TNet: FMRI to T1w MRI Knowledge Transfer Network for Brain Multi-phenotype Prediction." International Conference on Medical Image Computing and Computer-Assisted Intervention. Cham: Springer Nature Switzerland, 2024.
    - Fedorov, Alex, et al. "Self-supervised multimodal learning for group inferences from MRI data: Discovering disorder-relevant brain regions and multimodal links." NeuroImage 285 (2024): 120485.
    - Li, Tongtong, et al. "Automated diagnosis of major depressive disorder with multi-modal MRIs based on contrastive learning: a few-shot study." IEEE Transactions on Neural Systems and Rehabilitation Engineering (2024).

**Questions:**

- Please see weaknesses above.
- How were the confidence intervals computed?

---

### Official Review · Reviewer_X2ds · 2024-10-31

**Soundness:** 2
**Presentation:** 2
**Contribution:** 2
**Rating:** 5
**Confidence:** 3

**Summary:**

The authors have proposed an interesting contrastive pretraining framework that utilizes 3D T1 MRI and functional connectivity networks (FCNs) derived from fMRI data to learn robust representations for mental disorder diagnosis. The model has been evaluated in both linear probing and retrieval settings, showcasing intriguing ideas. However, the organization of the paper and the presentation of results could be enhanced.

**Strengths:**

1. The authors effectively describe the motivation behind this study and highlight the potential contributions of integrating contrastive pretraining with MRI and FCN data. The approach of cross-modal contrastive learning is well-founded and offers a solid framework for using information from both modalities to enhance diagnostic accuracy while also enabling retrieval/diagnosis when one of the modalities is missing.
2. The three objective functions are well-described, and the authors provide detailed ablation studies, which contribute to a better understanding of their impact on model performance.
3. The clear delineation between pretraining and evaluation sets facilitates relatively fair comparisons across out-of-domain datasets.

**Weaknesses:**

1. The writing could be improved to enhance the presentation of results. For instance, the experimental setup for network prompting is somewhat challenging to follow, as detailed in my questions below.
2. The authors hypothesize that the “CINP” model has potential for improvement through fine-tuning; it would be better to directly include corresponding results in the tables for a more comprehensive understanding.
3. The comparisons are primarily between CINP and single-modality models (sMRI or FCN). There is a lack of discussion and direct comparisons with existing multi-modal methods for mental health diagnosis, both in linear probing and fine-tuning contexts. At least, some consensus on FCN and SSP-based model predictions would allow for fairer comparisons.
4. Although several metrics are presented, the authors did not discuss in detail the differences, especially when two metrics offer contrasting results.
5. The authors did show the advantages of pretraining over simply fine-tuning a model directly on the evaluation dataset that utilizes both modalities as input. This is an important aspect to demonstrate the value of pretraining. Further improvements could include testing in a low-data regime to see if pretraining can reduce data requirements for subsequent fine-tuning.
6. While improvements are shown, the absolute values of metrics appear low for potential clinical applications. Providing context on results from the literature for the same task or similar datasets would help readers unfamiliar with this specific field better interpret the model's performance.

**Questions:**

1. Was the linear probing of CINP performed on concatenated MRI and FCN embeddings, or was it based on one of the modalities?
2. Regarding network prompting, what is meant by partitioning all samples into 5/10 subsets? What is the purpose of this partitioning? Additionally, could the authors clarify why they chose to use 10% or 50% of the data? Is this to assess retrieval performance in a low-data regime? It would also be helpful to explain why the results are better when using 10% of the training data compared to using 100%.Also, the number 29.33% seems inconsistent with the figures in Table 3.

---

### Author Response · Authors · 2024-11-28
**General Response**

We sincerely thank the reviewers for their valuable and constructive feedback, as well as their positive remarks on our paper. We are particularly grateful for the recognition of our work as **well-motivated** (@X2ds), presenting a **novel and interesting idea** (@Ay6P, X2ds), being **well-described and easy to understand** (@X2ds, nAjV), **showing good performance** (@nAjV, Rh9t), and **addressing a significant problem in mental disorder diagnosis** (@nAjV) that is both relevant and valuable to the ICLR community.

During the rebuttal process, we have thoroughly addressed the reviewers’ main concerns and provided additional experiments and clarifications to further strengthen our work. We are committed to resolving any remaining concerns and are open to additional questions or discussions to ensure clarity and comprehensiveness.

---

### Note · Authors · 2025-03-06

I have read and agree with the venue's withdrawal policy on behalf of myself and my co-authors.

---

### Meta-Review · Area_Chair_AoqB · 2024-12-17

**Metareview:**

The paper proposes CINP, a contrastive pre-training framework that combines 3D T1 MRI and functional connectivity networks (FCNs) from fMRI data to learn robust representations for diagnosing mental disorders. The authors demonstrate that the learned representations can be used with a linear probing protocol for classification. Reviewers acknowledged the relevance of the topic and found the integration of T1 MRI with FCNs an interesting direction.

However, the paper faces significant limitations. Reviewers raised concerns about the poor presentation in the Quantitative Results section, which makes it difficult to evaluate the claimed performance. Additionally, the technical contributions are perceived as limited, with only marginal improvements demonstrated. Although the authors engaged actively in discussions and provided further comparisons, the reviewers remained unconvinced. Given the unanimous recommendation for rejection, I also recommend rejection.

**Additional Comments On Reviewer Discussion:**

Additional Comments on Reviewer Discussion
The reviewers’ concerns, summarised with the authors' input, focus on the following key points:

Q1: Comparison with Multi-Modal Models
Reviewers suggested comparing CINP with multi-modal fusion methods. While the authors included comparisons with three multi-modal models (Cross-GNN, MMT, and CAMF) based on functional and structural MRI, these results were not extensively reviewed or confirmed by the reviewers. It would be better to refine and present these comparisons clearly in a future submission.

Q2: Limited Performance Improvement
The reviewers noted that the model's performance on the ABIDE dataset is not state-of-the-art (SOTA), and the limited improvement diminishes its potential for clinical applications. I agree that the observed gains are marginal, particularly when considering the practical clinical significance.

Q3: Comparison with Related Works
The reviewers suggested comparisons with other contrastive learning works integrating fMRI and sMRI. However, the authors did not provide this revision in the current draft, despite ICLR allowing reviewer engagement for further feedback.

Q4: Experimental Setting
Concerns were raised regarding the clarity of the dataset split and network settings. These concerns were not addressed in the revised draft, leaving ambiguities unresolved.

Q5: Results Analysis
Reviewers recommended providing detailed explanations of the metrics and findings related to diseases. While the authors expanded on performance under different settings (e.g., FCN outperforming sMRI), the conclusion that patterns associated with ASD are more pronounced in functional networks lacks meaningful clinical interpretation. What clinically relevant patterns emerge remains unclear.

In summary, while the paper has merit in exploring multi-modal pre-training for mental disorder diagnosis, it fails to address key concerns regarding clarity, performance improvement, and clinical relevance. Given the reviewers’ unanimous negative opinion, I also believe this paper is not yet ready for publication.

---

### Decision · Program_Chairs · 2025-01-22

Reject